# Variational Inference with Unnormalized Priors

## Abstract

Variational inference typically assumes normalized priors, limiting the expressiveness of generative models like Variational Autoencoders (VAEs). In this work, we propose a novel approach by replacing the prior $p(z)$ with an unnormalized energy-based distribution $\exp\left(-E(z)\right)/Z$, where $E(z)$ is the energy function and $Z$ is the partition function. This leads to a variational lower bound that allows for two key innovations: (1) the incorporation of more powerful, flexible priors into the VAE framework, resulting in improved likelihood estimates and enhanced generative performance, and (2) the ability to train energy-based models (EBMs) without the need for computationally expensive Markov chain sampling, requiring only a small $n > 1$ importance samples from the posterior distribution. Our approach bridges VAEs and EBMs, providing a scalable and efficient framework for leveraging unnormalized priors in probabilistic models.

## 1 Introduction

Generative models are essential in unsupervised learning and data generation, with each approach offering unique strengths and facing specific challenges. Among these, Variational Autoencoders (Kingma & Welling, 2022), normalizing flows (Rezende & Mohamed, 2016; Kingma & Dhariwal, 2018), score-based/diffusion models (Song & Ermon, 2020; Sohl-Dickstein et al., 2015; Ho et al., 2020), and energy-based models (Du & Mordatch, 2020; Grathwohl et al., 2020) represent some of the most influential methods in modern generative modeling. Each of these models brings distinct advantages but also limitations that impact their practical application and effectiveness.

Variational Autoencoders (VAEs) are a cornerstone in generative modeling due to their efficiency and scalability. VAEs utilize the variational lower bound (VLB) to approximate complex posterior distributions and have demonstrated considerable success in various tasks such as image generation (Vahdat & Kautz, 2021; Child, 2021) and anomaly detection (Pol et al., 2020). The primary strength of VAEs lies in their ability to efficiently model large datasets through a combination of variational inference and neural network architectures. However, VAEs face a significant challenge due to their use of simple, normalized priors, such as Gaussian distributions. This simplicity can lead to a misalignment between the prior and the posterior, where the model struggles to capture the true complexity and multi-modality of the data. Although efforts to enhance the flexibility of the posterior have been made (Rezende & Mohamed, 2016; Kingma et al., 2017), these methods do not fully resolve other issues pertaining to quality image generation (Dai & Wipf, 2019).

Normalizing flows offer an alternative by applying a series of invertible transformations to a base distribution, allowing for the modeling of complex data distributions with exact likelihood computation. This flexibility makes normalizing flows highly expressive compared to VAEs. However, the challenge lies in designing and training these transformations, which can become computationally demanding and complex, particularly as the dimensionality of the data increases. As a result, while normalizing flows provide powerful modeling capabilities, they may not always be practical for large-scale or real-time applications.

Score-based models and diffusion models (SDMs) represent another innovative approach by learning to model the score function, or the gradient of the log-likelihood, of the data distribution. These models refine noisy data through iterative denoising, leading to high-quality samples and the ability to model intricate data structures. Despite their impressive performance, SDMs face substantial training and sampling challenges. Training involves optimizing the score function across multiple

noise levels, which requires extensive computation. Additionally, the sampling process is typically slow, as generating high-quality samples often involves many iterative refinement steps. These factors can limit the practicality of SDMs for large-scale or real-time generative tasks.

Energy-based models (EBMs) offer a different paradigm by defining probability distributions through an unnormalized energy function. EBMs can capture highly complex and varied data distributions due to their flexible energy function. However, the practical application of EBMs is constrained by the need for computationally intensive sampling methods like Markov Chain Monte Carlo (MCMC), which are necessary to approximate the intractable partition function. This reliance on expensive sampling techniques makes EBMs less scalable and efficient compared to other generative models.

Among these approaches, Variational Autoencoders (VAEs) remain our primary focus due to their foundational role in generative modeling and their widespread application in various domains. The core limitation of VAEs lies in their posterior parameterization failing to effectively capture the complexity of the prior distribution. While enhancing the flexibility of the posterior has been explored, this does not fully capture the data distribution.

In this work, we address this limitation by introducing unnormalized energy-based priors into the VAE framework. By incorporating flexible, unnormalized priors, we aim to improve the alignment between the prior, posterior, and even the reconstruction likelihood. This novel approach leverages the expressiveness of energy-based models while maintaining the computational efficiency of VAEs. Our method provides a scalable solution that enhances generative performance and likelihood estimation, positioning unnormalized priors as a powerful tool for advancing VAE capabilities and addressing their core limitations.

## 2 LIKELIHOOD ESTIMATOR FOR UNNORMALIZED PRIORS

Consider the following formulation of the variational lower bound:

$$\ln p(x) \geq \mathop{\mathbb{E}}_{q(z|x)} [\ln p(x|z) + \ln p(z) - \ln q(z|x)] \tag{1}$$

Where $\ln p(x|z)$ is the reconstruction likelihood, $\ln p(z)$ is the prior and $\ln q(z|x)$ is the approximate posterior. We can represent the prior $p(z)$ in terms of a Boltzmann distribution $\exp(-E(z))/Z$, where $E(z)$ is the energy function and $Z = \int \exp(-E(z))dz$ is the partition function or normalizing constant. The VLB then becomes:

$$\ln p(x) \geq \mathop{\mathbb{E}}_{q(z|x)} [\ln p(x|z) - E(z) - \ln q(z|x)] - \ln Z \tag{2}$$

The main issue here pertains to the partition function as it is generally intractable to compute. When training pure energy-based models, samples from the model are required to be generated during training to approximate its gradient, which is a difficult endeavour in and of itself as it requires a high quality sampler. For our purposes, we instead exploit the approximate posterior to our advantage to estimate the partition function through self-normalized importance samples, leading to the following biased but consistent estimator of the VLB:

$$\ln p(x) \geq \mathop{\mathbb{E}}_{q(z|x)} [\ln p(x|z) - E(z) - \ln q(z|x)] - \ln(\mathop{\mathbb{E}}_{q(z|x)} [\exp(-E(z) - \ln q(z|x))]) \tag{3}$$

Unlike pure EBMs with which likelihood computation is intractable and training requires expensive Markov chain sampling, the EBM prior can be approximated in an unbiased fashion with Monte Carlo samples from the approximate posterior. In effect, the unnormalized prior VLB gives us a framework with which EBMs can be trained much more efficiently and within a rigorously justified maximum-likelihood framework. Thanks to the generality of this VLB, the choice of $E(z)$ can be arbitrary, ranging from simple restricted Boltzmann machines to large ResNets for higher-dimensional datasets. For simplicity, we will be focusing only on Gaussian-Bernoulli RBMs as the energy prior for the remainder of the paper. The Gaussian-Bernoulli RBM is a specific formulation

of restricted Boltzmann machines in which the visible units parameterize a Gaussian distribution, while the hidden units parameterize a Bernoulli distribution, realizing a universal approximator of mixture models (Krause et al., 2013; Gu et al., 2022). The marginal energy of a Gaussian-Bernoulli RBM is as follows (Liao et al., 2022):

$$E(z) = \frac{1}{2}(\frac{z-\mu}{\sigma})^\top(\frac{z-\mu}{\sigma}) - \text{Softplus}(W^\top\frac{z}{\sigma^2} + b)^\top\mathbf{1} \tag{4}$$

Where $\mu$ and $\sigma$ are the per-visible unit mean and standard deviation vectors, $b$ is the hidden bias vector and $W$ is the weight matrix of the RBM.

After model training, VAEs are often also evaluated using the importance-sampled negative log-likelihood, which has a tighter bound over the VLB. We may compute this also using self-normalized importance sampling as such:

$$\ln p(x) = \ln \sum_z \exp(\ln p(x|z) - E(z) - \ln q(z|x)) - \ln \sum_z \exp(-E(z) - \ln q(z|x)) \tag{5}$$

Although not explored in this work, one can also train energy-based importance-weighted autoencoders (Burda et al., 2016) by directly optimizing the above bound.

## 3 RELATED WORK

Incorporating flexible priors such as energy-based, score/diffusion-based, and mixture priors into variational autoencoders (Vahdat et al., 2021; Han et al., 2020; Lee et al., 2023; Rombach et al., 2022) and also regular autoencoders (Ghosh et al., 2020; Jing et al., 2020) is not a new concept, and has seen some considerable success. However, these attempts have approached this concept from a fundamentally different perspective that divorces the objective from its probabilistic roots, resulting in what is essentially just a "packaging" of two different models. This can have potentially suboptimal effects, espcially for regular autoencoders due to the inherent nature of their structure. Regular autoencoders, unlike VAEs, lack the probabilistic underpinnings that would enable them to effectively handle complex priors. This is because regular autoencoders rely on a deterministic encoder, where the posterior $q(z|x)$ can be considered degenerate. As a result, attempts to enhance the prior distribution, often through ex-post density (Ghosh et al., 2020; Jing et al., 2020) modeling techniques, have not yielded substantial improvements.

One key reason for this limitation is the issue of disjoint and discontinuous energy landscapes in regular autoencoders when incorporating sophisticated priors like energy-based models. Without the probabilistic backbone of variational inference, the latent space in regular autoencoders can become highly irregular, leading to poor generalization. In practice, this often results in samples that contain significant artifacts, as the model struggles to reconcile the discontinuities in the energy landscape. These artifacts are a consequence of the model's inability to effectively smooth the transitions between different modes in the data distribution, a problem exacerbated by the lack of a robust posterior to regulate the latent space.

In contrast, our proposed method directly addresses this issue by unifying these flexible priors into the rigorous framework of variational inference. By optimizing the variational lower bound in the presence of an unnormalized energy-based prior, our approach ensures that the latent space remains well-structured and continuous. Specifically, the VLB in our framework is expressed as in Equation (3).

For a single-sample approximation, the VLB reduces to $\ln p(x|z)$, which is exactly the regular autoencoder objective. In this context, regular autoencoders can viewed as having a degenerate, deterministic posterior which fails to fully capture the complexity of the latent space.

Our proposed method of incorporating unnormalized, flexible priors into the VAE framework is orthogonal to the use of normalizing flows for improving posterior estimates (Rezende & Mohamed, 2016; Grathwohl et al., 2018). While both strategies aim to make models more expressive, they address considerably different limitations.

Normalizing flows focus on improving the posterior distribution $q(z|x)$ by applying a series of invertible transformations to a simple base distribution (typically Gaussian). These transformations introduce more flexibility into the posterior, allowing it to better approximate the true latent distribution. However, normalizing flows are subject to important design constraints. In order to ensure computational efficiency, the transformations applied in normalizing flows must remain computationally cheap, which places a natural limit on how complex or expressive the posterior can be. Although they are more flexible than traditional Gaussian posteriors, NFs may not fully capture the complexity of highly multi-modal or intricate data distributions due to these computational constraints.

On the other hand, our approach addresses the prior distribution $p(z)$ rather than the posterior. By introducing energy-based priors, we make the prior more flexible and capable of capturing complex latent structures, leading to better alignment with the posterior. Learning a more expressive prior helps mitigate the limitations of normalizing flows, which might not be fully expressive on their own due to the aforementioned computational trade-offs. Importantly, the use of unnormalized priors ensures that the model remains efficient, while simultaneously providing it with a richer latent space.

Another similar approach to normalizing flows is adversarial Variational Bayes (Mescheder et al., 2018), where the posterior is matched to prior *implicitly* via an adversarial objective. AVB is the closest to our approach, in that despite the implicitness, the resulting objective is still a valid variational lower bound. Unlike our energy-based approach, the AVB objective is inherently unstable, since the density-matched KL estimate is an adversarial objective which are known to be troublesome to train. On the other hand, the auxiliary model in our energy-based objective is the model's prior, making the training objective complimentary and stable.

# 4 OPTIMIZATION DIFFICULTIES

Scaling VAEs to larger datasets, such as MNIST, comes with optimization difficulties due to the emphasis on maximizing likelihood during training. This often leads to compromised generative performance as generative performance is neither sufficient nor necessary for good likelihoods (Theis et al., 2016).

More specifically, the KL divergence between the prior and posterior becomes high, indicative of mismatch or latent-space "holes". Sampling from these holes, which are areas of low energy, will result in samples that contain noticable artifacts (Rezende & Viola, 2018). This often happens when the prior is fixed, causing the model to trade-off between quality samples (which require a flexible latent-space) and learning the perfect density (leading to posterior collapse).

A straight-forward way to work around this issue is to make the prior learnable, and this is a legitimate solution that can largely mitigate this issue. However, the simultaneous training of multiple objectives may lead to unexpected behaviour such as an overpowered decoder that the prior has difficulty catching up to. Moreover, in hierarchical models (Vahdat & Kautz, 2021; Hazami et al., 2022) the optimization can become very unstable.

Instead, we use ex-post density estimation (XPDE) (Ghosh et al., 2020) to correct prior-posterior mismatch. XPDE works because the density estimator models the aggregated posterior, which is the empirical prior that the approximate posterior encompasses. The aggregated posterior is in fact the optimal solution (Tomczak & Welling, 2018), resulting in a KL of zero. For this reason, we use the log-likelihood of the XPDE prior in place of the original energy-based prior for model evaluation. The energy prior is still very much useful as a latent-space regularizer, preventing the latent-space from becoming too disjoint and thus difficult to capture while allowing for much better log-likelihoods than the equivalent Gaussian VAE. We specifically use a Gaussian mixture model, although more "serious" datasets would use more sophisticated approaches like two-stage VAE training (Dai & Wipf, 2019), which combines ex-post density estimation with hierarchical modelling.

## 5 EXPERIMENTS

We demonstrate the validity and effectiveness of the unnormalized prior VLB through density estimation on both toy data as well as real data.

### 5.1 TOY 2D DATA

We first put to test the energy-based VAEs on a series of 2D toy data. In particular, we use the 8-Gaussians, checkerboard and 2-spirals from (Cao et al., 2019) as well as the four potentials from (Rezende & Mohamed, 2016).

We test two different variations of the energy-based VAEs on each benchmark; one with an unconstrained (learnable variance) Gaussian posterior (EVAE), and one with a fixed-variance ($\sigma = 1$) Gaussian posterior (ECVVAE). We do this to demonstrate how flexible posteriors can actually hurt the learning of the energy-based prior as it will attempt to learn a degenerate distribution. We also test a standard VAE with an isotropic Gaussian prior and factorized Gaussian posterior for reference (VAE).

All of the VAEs tested have the same architecture of fully-connected DenseNet (Huang et al., 2018) blocks with hidden size of 16 units and depth size of 4 in both the encoder and decoder. The two energy-based VAEs have a Gaussian-Bernoulli RBM prior with 16 visible units and 32 hidden units. Weight normalization (Salimans & Kingma, 2016) is applied to all of the layers in the encoder and decoder. We use the Gibbs-Langevin sampler Liao et al. (2022) to sample from the energy VAEs.

RESULTS

The learned distributions of the VAE and our energy VAEs can be seen in Table 1. For the datasets on the left, the VAE clearly has trouble learning the correct density for most of the data, while the unrestricted EVAE outright fails due to it degenerating. The ECVVAE on the other hand displays much better behaviour, successfully capturing two of the four datasets and showing a good attempt at capturing the other two distributions. Similar behaviour is seen for the energy potentials, where the regular VAE has trouble assigning energies correctly, and the ECVVAE displaying relatively better mode coverage.

| GT | VAE | EVAE | ECVVAE | GT | VAE | EVAE | ECVVAE |
|----|-----|------|--------|----|-----|------|--------|

Table 1: Left: density estimation on toy 2D data (top three are from (Cao et al., 2019)). Right: density estimation on four potentials from Table 1 of Rezende & Mohamed (2016)

### 5.2 MNIST

To demonstrate its ability to scale to higher dimensions, we train a two-stage fully-connected ECVVAE of layers 784-256-64 in the encoder and 64-256-784 in the decoder on dynamically binarized MNIST. The model is trained across three runs, and the training log is shown in Figure 1. Samples from the energy prior, generated using block Gibbs sampling, are shown in Figure 2. Samples from the GMM priors are shown in Figure 3.

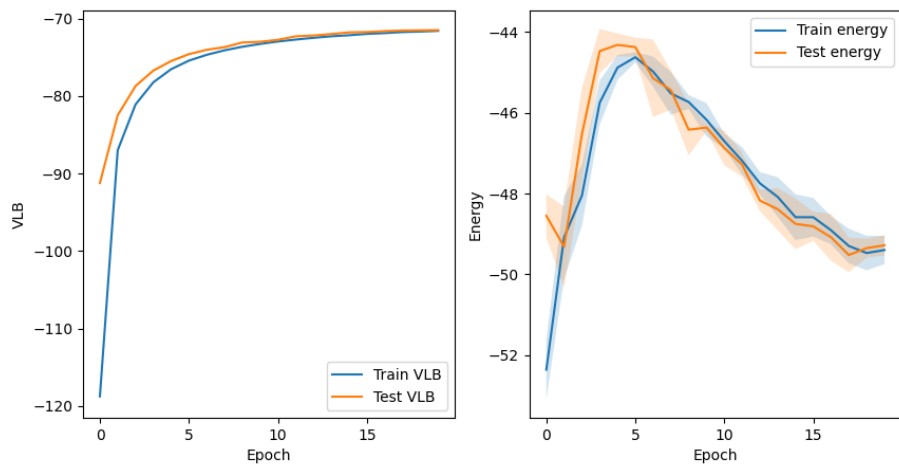

Figure 1: Left: variational lower bound across training iterations. Right: prior energy across training iterations.

RESULTS

As descibed earlier, MNIST was much more difficult to model than the toy datasets, with the model invariably preferring to maximize reconstruction error at the expense of the energy-function which increases throughout training. This is the case even when incorporating design changes that discourage it, such as the aforementioned variance fixing.

We also experimented with both volume-preserving and non volume-preserving inverse autoregressive flows in the posterior to see if they provided meaningful performance gains. Neither model provided any meaningful gains, possibly due to the fact that both approaches are not very different in practice Kingma et al. (2017).

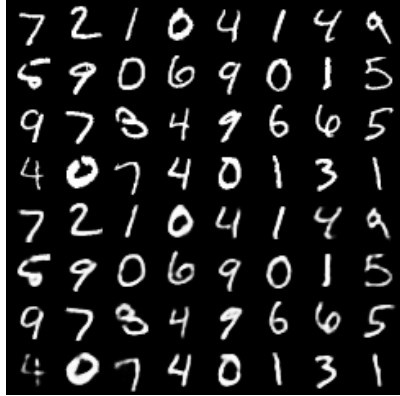 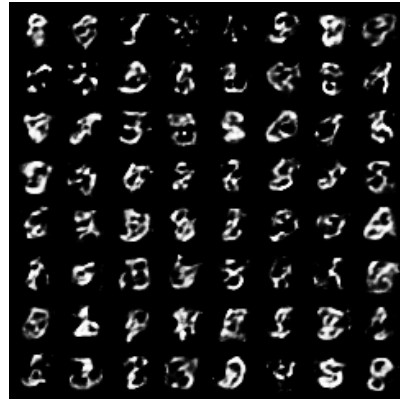

Figure 2: Left: Test-set reconstructions from the ECVVAE. Right: Samples from ECVVAE (w/ 5 Gibbs steps).

Once again, as sample-generation and log-likelihoods are not mutually exclusive, the state-of-the-art log-likelihoods attained on MNIST by the single-stage model (see Table 2) is completely justified even though the samples are not remotely close to it. Similarly, the GMM samples are much better, but they come with worse likelihood estimates.

Unnormalized samplers can still be useful though, especially for latent-space interpolation where intermediate samples situated in low energy density regions can be corrected (Creswell et al., 2017; Creswell & Bharath, 2018).

| Model | $\approx p(x)$ |
|---|---|
| NVAE w/o flow (Vahdat & Kautz, 2021) | 78.01 |
| IAF-VAE Kingma & Dhariwal (2018) | 79.10 |
| CR-NVAE (Sinha & Dieng, 2022) | 76.93 |
| BFN Graves et al. (2024) | 77.87 |
| ECVVAE | **69.36** |
| ECVVAE w/ 4-comp GMM | 161.86 |
| ECVVAE w/ 10-comp GMM | 154.54 |

Table 2: Comparison on binarized MNIST, test set average negative log likelihood (lower is better).

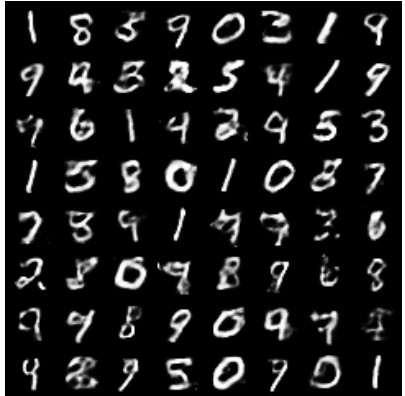

Figure 3: Left: Samples generated from a 4-component GMM. Right: Samples generated from 10-component GMM.

## 6 CONCLUSION

In this paper, we proposed a novel variational inference framework that integrates unnormalized energy-based priors into the Variational Autoencoder (VAE) model. By replacing the traditional normalized prior with a more flexible energy-based distribution, we addressed key limitations of VAEs, particularly their inability to model complex, multimodal data distributions. Our method demonstrated both theoretical and practical advantages, including improved likelihood estimation and generative performance, as well as scalable training of energy-based models without relying on expensive Markov Chain Monte Carlo (MCMC) sampling. We empirically validated our approach on both toy and real-world datasets, showing that energy-based VAEs (EVAEs) outperform traditional VAEs in terms of capturing complex data distributions and producing high-quality generative models. Although our experiments primarily focused on Gaussian-Bernoulli RBM priors, the framework is versatile and can be applied to a wide range of unnormalized priors.

## 7 DISCUSSION

The introduction of unnormalized priors into VAEs offers a new perspective on generative modeling by bridging the gap between VAEs and energy-based models (EBMs). Unlike prior work that seeks to enhance generative models by combining different techniques without a unified probabilistic foundation, our approach maintains the rigor of variational inference. This not only ensures the tractability of likelihood-based training but also leverages the expressiveness of energy-based models to enrich the latent space of the VAE. By doing so, we have effectively addressed one of the core limitations of VAEs—namely, the mismatch between simple priors and complex posterior distributions.

Our experiments on toy datasets highlight the capability of our model to better capture multimodal and intricate data distributions compared to standard VAEs. While the EVAE variant struggled due to its flexibility, the ECVVAE, which constrained the posterior variance, demonstrated superior performance, particularly in learning more robust and accurate latent representations. The results

suggest that the interplay between posterior flexibility and energy-based priors must be carefully balanced to avoid degenerate solutions, such as overfitting to posterior variances.

When scaling to more complex datasets like MNIST, our method faced challenges in maintaining a balance between the reconstruction objective and energy minimization, particularly in high-dimensional settings. Nonetheless, the experiments showed that our energy-based VAEs still achieved state-of-the-art log-likelihood results, affirming the potential of unnormalized priors for large-scale generative modeling tasks. Further exploration of architectural design, variance control, and flexible posteriors could help refine the model's ability to handle such datasets more effectively.

Future work could explore alternative strategies for posterior optimization, including hybrid approaches that combine energy-based priors with more expressive posterior distributions like normalizing flows or score-based methods. Using ECVVAEs as hierarchical priors could also potentially improve the representation of complex data structures by conserving energy across layers. Additionally, applying this framework to more diverse and complex datasets would provide further insights into its generalizability and performance across various domains.

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
