# OpenReview forum: "Variational Inference with Unnormalized Priors"
_ICLR.cc/2025/Conference — ICLR 2025 Conference Withdrawn Submission_

### Official Review · Reviewer_ftj4 · 2024-10-24

**Soundness:** 1
**Presentation:** 1
**Contribution:** 1
**Rating:** 3
**Confidence:** 5

**Summary:**

The authors propose an energy-based prior in VAE. It is considered to by more flexible and efficient in training.

**Strengths:**

* The energy-based prior might be a possible choice to be considered in VAE.

**Weaknesses:**

* There are some technically problematic sentences, for example:
    * Line 033: VLB is not to approximate posterior but the log marginal $\ln p(x)$
    * Line 066: "The core limitation of VAEs lies in their posterior parameterization failing to effectively capture the complexity of the prior distribution". This sentence is confusing
    * It is common to use ELBO rather than VLB.
    * Line 138: "Regular autoencoders, unlike VAEs, lack the probabilistic underpinnings that would enable them to effectively handle complex priors." This might not be correct. VAE is just a model, no matter what prior you choose, it is still a VAE. Variational inference (VI) is just a commonly used approach to solving VAE. VI may fail with some complicated non-Gaussian priors, but there are still a lot of inference/learning methods that can handle non-Gaussian complicated priors, e.g., https://proceedings.neurips.cc/paper/2017/hash/35464c848f410e55a13bb9d78e7fddd0-Abstract.html, https://openreview.net/pdf?id=HD5Y7M8Xdk, etc.
    * Line 156 is wrong.
    * What does line 192 mean?
* There is no equation in Sec. 4, so it is very hard to understand claims in Sec. 4.
* The paper also considers their posterior $q$ to be flexible, but there is no detailed explanation of what is their $q$. Actually, there are two posteriors, the posterior from the generative model we want to approximate $p(z|x)$, and the approximated posterior $q(z|x)$ which is the variational distribution or encoder. It is very hard to distinguish those in this paper.
* The experiment is too simple and hard to understand.
* There is no detailed explanation why it is easy to train. Since the prior is not a sequential formula, it is always easy to sample.

**Questions:**

/

---

### Official Review · Reviewer_eB8v · 2024-10-29

**Soundness:** 2
**Presentation:** 1
**Contribution:** 1
**Rating:** 3
**Confidence:** 4

**Summary:**

This paper explores the following research question: Can we specify un-normalized priors in variational inference? A positive answer to this question would mean more expressive power in models like VAEs, since one could create more expressive priors by specify an energy function $E(z)$ leading to $\exp(-E(z)) / Z$ as the prior (with unknown partition function $Z$), instead of a restricted normalized $p(z)$ like the standard gaussian.

The paper claims to show a positive answer to this research question. In section 2, the authors claim to show an adjustment (in Eq (3)) to the standard ELBO (in Eq (1)) that utilizes samples from the posterior $q(z|x)$ to approximate the term involving the unknown $Z$. In section 5, the authors attempt to empirically demonstrate improvements by using the claimed theoretical adjustment, by running experiments on a synthetic 2D dataset and the MNIST dataset.

**Strengths:**

- The paper asks a relevant research question. If it can be shown conclusively that VAEs can be trained with more flexible priors (like an energy based function), it would certainly advance our understanding of variational models and potentially improve the state-of-the-art.
- The synthetic experiments (Table 1) clearly demonstrate that this approach could be promising.

**Weaknesses:**

**Technical Weaknesses**:
- *The theoretical claims in section 2 are not well-substantiated.* It is claimed in lines 95-97 that eq(3) leads to a biased but consistent estimator of the VLB (Variational Lower Bound, or the ELBO). Neither the biased-ness nor the consistency of this estimator are shown in any form. Much of the paper relies on this equation.
- *Limited breadth of experiments.* Apart from the synthetic dataset, the only dataset shown is MNIST. The breadth of experiments is not enough to establish the claim of improvements from using an unnormalized prior.
- *Unclear interpretation from the experiments shown.* Even in the MNIST experiment shown, some of the takeaways are unclear. In particular, what is the takeaway from Figure 1? It is unclear from lines 267-269 where that figure is mentioned.
- *Limited explanation for certain design choices and phenomena.* For example -- (1) Why were the VAE dimensions for the MNIST dataset chosen as mentioned in line 267? (2) Why does ECVVAE (with the variance constraint) outperform EVAE (without the variance constraint, i.e. more flexible)? It is mentioned in lines 240-242 that EVAE can lead to a degenerate solution, but it would have been nice to substantiate that claim with evidence.

Overall, I found that both the theoretical arguments and the experimental evidence are not enough to support the claims of the paper.

**Presentation Weaknesses**: Below I provide some examples of the writing being imprecise or unclear.
- In eq (5), it is not clarified what is the domain of the summation $\sum_{z}$
- Line 154, there is no link to equation (3).
- Lines 134-137 are hard to follow. What does "divorces the objective from its probabilistic roots" mean? It seems like some parts of the paper are written like a position paper.

**Questions:**

See weaknesses above.

---

### Official Review · Reviewer_7Qcr · 2024-10-31

**Soundness:** 2
**Presentation:** 1
**Contribution:** 1
**Rating:** 3
**Confidence:** 4

**Summary:**

In the submitted manuscript, authors propose to use energy-based models as priors in Variational Autoencoders (VAEs).  They claim that with this they resolve some of the main limitations of VAEs, however, it is not clear to me which limitations exactly they address. They test out their approach on synthetic data as well as MNIST dataset, however, they fail to compare to a lot of related literature (e.g., in their first experiment they compare only with a standard VAE with an isotropic Gaussian prior)

**Strengths:**

- The idea of combining VAEs and energy based models is interesting

**Weaknesses:**

- The paper is written badly. For example, in the introduction the authors mainly try to explain their focus on VAEs (as opposed to other generative models like flows or diffusion), and they only briefly describe their contribution in the last 1 or 2 paragraphs. Also, the methods section could be expanded, it is not entirely clear to me how you go from Eq. (2) to Eq. (3)
- Moreover, there are a lot of statements that I find confusing and that should be elaborated more on. Could the authors provide further explanations for the following statements
    - lines 66-67: *The core limitation of VAEs lies in their posterior parameterization failing to effectively capture the complexity of the prior distribution.* Are you talking about the posterior collapse problem here, or?
    - lines 134-136: *However, these attempts have approached this concept from a fundamentally different perspective that divorces the objective from its probabilistic roots, resulting in what is essentially just a ”packaging” of two different models.* Why does previous work on priors in VAEs "divorces the objective from its probabilistic roots"?

- From what I understand, there has been a large body of work on integrating more complex priors into VAEs [1, 2, 3]. No proper connection to the prior literature on priors in VAEs is made in this work. After reading the manuscript, it is not clear to me what their approach enables compared to existing literature (both from the theoretical perspective as well as empirical)
- There has even been prior work on integrating energy based models into VAEs [4], that is again unaddressed in the manuscript
- It seems to me that introduction of energy based models introduces scalability issues. Is that the reason why authors only test out their approach on MNIST? If that's the case, then I fail to understand what is the added benefit of introducing energy based models
- I do not find experimental results convincing. I do not think the paper is strong/interesting enough theoretically that it would justify only MNIST-level experiments

[1] [Hyperspherical Variational Auto-Encoders](https://arxiv.org/pdf/1804.00891)

[2] [Gaussian Process Prior Variational Autoencoders](https://arxiv.org/abs/1810.11738)

[3] [Mixture-of-Experts Variational Autoencoder for Clustering and Generating from Similarity-Based Representations on Single Cell Data](https://arxiv.org/abs/1910.07763)

[4] [A Contrastive Learning Approach for Training Variational Autoencoder Priors](https://arxiv.org/pdf/2010.02917)

**Questions:**

See weaknesses above

---

### Official Review · Reviewer_KMNc · 2024-11-03

**Soundness:** 3
**Presentation:** 2
**Contribution:** 2
**Rating:** 3
**Confidence:** 4

**Summary:**

This paper considers models deep generative models comprising an energy model for the prior over latent variables p(z) and then the latents are mapped to the observations x using a neural network which parameterises p(x|z). Variational inference is used to handle the intractability in the likelihood functions as is the case for VAEs, introducing an approximate posterior distribution. The intractability arising from the normalisation constant of the energy-model prior is handled by (self-normalised) importance sampling using the approximate posterior as the proposal distribution. The marginal of a Gaussian-Bernoulli Restricted Boltzmann Machine (RBM) is used for the energy model prior. The approach is tested on simple 2D synthetic examples and dynamically binarised MNIST.

**Strengths:**

The writing is clear.

Deep generative models with complex priors is an interesting direction of work.

I like the creative combination of variational inference, importance sampling, VAEs and energy based models.

The results on the simple 2D densities are promising. (Although lots of methods look good in these simple cases e.g. flows.)

**Weaknesses:**

In general, this work felt like it was a bit early to be published and needed more work spelling out important details and extending the experimental comparisons.

Many details are missing for the MNIST experiment e.g. the precise models used for the experiments such as p(x|z) mentioned above, the number of visible and hidden units, number of parameters, the optimiser and hyper-parameters etc.

More generally, I would have liked an algorithm box / pseudo-code describing the algorithm in one place, noting the stochastic approximations, Monte Carlo sampling etc.

I would have liked a short description of ex-post density estimation as this is key for the results. I would have liked more detail on how it was used in the MNIST experiments - what density estimator is used?

I agree completely that good log-likelihoods do not necessarily mean better samples, but for MNIST the samples look very poor whereas the held-out log likelihoods are state-of-the-art. This is surprising. It is possible to get excellent looking samples with terrible likelihoods, e.g. via over-fitting, but to see the reverse is very unusual. Is it possible that you are modelling discrete data with a continuous density model? This can inflate the numbers as the density can put delta function spikes down. It wasn’t clear what the probabilistic model is for the MNIST experiment e.g. what the likelihood p(x|z) is.

The method is only evaluated on synthetic data sets and MNIST. This is no longer sufficient in my view and there needs to be more datasets used for evaluation and these datasets need to be more challenging.  E.g. taking the models cited in table 2 as a guide for the rough level of experimentation that is typically performed we have
IAF-VAE - various versions of MNIST, CIFAR10
NVAE - MNIST, CIFAR-10, CelebA 64, and CelebA HQ
CR-NVAE - MNIST, CIFAR-10, and CelebA
BFN - MNIST and CIFAR-10, Language Modelling Tasks

The above point is particularly key in order to substantiate the claim that the method produces state-of-the-art probabilities. I would also suggest verifying the quality of the model by another independent task e.g. missing data imputation, to verify that it really is performing well.

**Questions:**

Importance sampling from a variational proposal is generally thought to be a bad idea e.g. see MacKay, Information Theory, Inference, and Learning Algorithms Exercise 33.6 (variational distributions are more compact to than the true posterior so importance sampling will lead to very high variance in the importance weights and low effective sample sizes). Why is this not an issue here?

It would be useful to have an explanation for why EVAE underfits so much more than ECVVAE (e.g in table 1, the EVA). The following paper might help explain it: Two problems with variational expectation maximisation for time-series models, Richard Eric Turner and Maneesh Sahani. This contains examples where more expressive q distributions perform worse and provides an explanation in terms of the parameter-dependent tightness of the ELBO.

What is the "prior energy” shown in Fig 1? Is this the average energy over both training samples and datapoints? Could this be clarified?

The text says "the energy-function which increases throughout training”, but the figure does not show monotonic increases in energy. Could you explain what is meant / plotted here?
line 300 and Table 2 — there are some \cites which should be \citeps

---

### Official Review · Reviewer_NiwF · 2024-11-04

**Soundness:** 2
**Presentation:** 2
**Contribution:** 1
**Rating:** 3
**Confidence:** 4

**Summary:**

The paper proposes a bound for variational inference (more precisely, variational expectation maximization) with unnormalized (energy-based) learned priors.
The problem in this setup is that a computationally infeasible prior partition function $Z$ appears in the ELBO.
The paper proposes to estimate $Z$ by importance sampling during training.
The method seems to suffer from "optimization difficulties" when scaling to moderately large setups (MNIST), and the authors propose to replace the learned prior with other density estimates once training is done.

**Strengths:**

The problem statement is clear, and derivations are easy to follow.
The discussion does not oversell the contribution nor the complexity of the proposal.

**Weaknesses:**

I believe that, in estimating the partition function $Z$ by importance sampling (Eq. 3), the paper misses a crucial point of (black-box) variational inference (VI).
This concern would explain the "optimization difficulties" that deserved a dedicated section (Section 4), and it would likely prevent the proposed method to scale.

I will elaborate in detail below.
In a nutshell, my argument is that, especially in the context of variational expectation maximization (e.g., VAEs, on which the paper focuses), the entire point of VI is to avoid the high variance of importance sampling.
However, the proposal in Eq. 3 brings importance sampling back.
At this point, one could arguably as well consider training the entire model by importance sampling (at least I don't see an obvious reason why this would make things substantially worse at this point).

Additionally (and possibly even more severely), while the right-hand side of Eq. 3 is indeed a lower bound on the log marginal likelihood, I don't think the expectation of its estimator would be.
I can't find an explicit discussion of how the bound is estimated in practice, but I assume that both expectations in Eq. 3 are estimated by sampling from $q$.
Due to the logarithm around the second term and the negative sign, I would expect the _expectation of the estimator_ of this second term is in fact an _upper_ bound on $-\ln Z$.
Thus, it is no longer clear whether the estimated objective as a whole is upper bounded in expectation, and this could lead to catastrophic issues when maximizing it.

## Detailed Argument

Different to the initial works on VI (which evaluate the ELBO analytically in specific models where the corresponding integrals can be solved), _black-box_ VI (the dominant variant of VI these days) estimates the ELBO by sampling from $q$.
Especially in the context of variational expectation maximization, where the goal is to _maximize_ the marginal likelihood over model parameters, this raises the question: if we have to estimate the objective anyway, why do we still settle for a (biased) bound on the marginal likelihood when we could as well be estimating the marginal likelihood itself?

- The ELBO is:
  $\ln p(x) \geq \mathbb{E}_{q(z|x)}\left[\ln\left(\frac{p(z, x)}{q(z|x)}\right)\right]$,
- whereas the true log marginal likelihood has a very similar form:
  $\ln p(x) = \ln\left(\mathbb{E}_{q(z|x)}\left[\frac{p(z, x)}{q(z|x)}\right]\right)$.

So both can be estimated by sampling from $q(z|x)$, but the ELBO introduces an additional bias by pulling the logarithm inside.
So why, in the age of black-box estimation, do we still maximize the ELBO?
Why don't we instead maximize the true marginal likelihood, which doesn't introduce a bias, and which can seemingly be estimated in a very similar way?

The reason why we don't simply estimate the marginal likelihood by importance sampling using the second formula above is that the importance weight $p(z,x)/q(z|x)$ has extremely high variance (roughly exponentially in the dimension of $z$ [1]).
This high variance gets inherited by the gradient estimate, rendering expectation maximization by importance sampling completely hopeless in even moderately high-dimensional latent spaces.
By pulling the logarithm inside the expectation, VI reduces the variance significantly, at the cost of introducing a bias.
Thus, black-box VI can be seen as _biased importance sampling_ [1].
In fact, different approaches to VI such as $\alpha$-VI [2] or importance weighted VI [3] can be understood as simply making different trade-offs on the bias-vs-variance scale, all with the goal of avoiding the prohibitively high variance that importance sampling would entail.

The present paper, however, seems to be taking the reverse route.
It starts from VI (i.e., biased importance sampling), encounters the problem of a computationally infeasible partition function, and then proposes to revert to normal importance sampling.
It is unclear to me why this approach should be significantly better than just staying with importance sampling (for the entire marginal likelihood) from the beginning, and I am not surprised that the authors ran into significant optimization problems, and that they had to limit experiments to relatively small setups (toy data + MNIST).

It is true that the proposal in Eq. 3 does not perform importance sampling for the entire bound, and it could turn out that the proposed choice of restricting importance sampling only to the partition function of the prior and doing regularized importance sampling (aka variational inference) for the rest of the model might lead to lower gradient variance for some (unknown) reason.
But if this is the case, then I would consider it as "accidental" since the choice of Eq. 3 does not take these considerations into account (so even if it did turn out to work better, we wouldn't understand why).

## Suggestion

As an alternative to estimating $Z$ by importance sampling, I would suggest to look into the literature for _upper_ bounds on importance sampling.
Since $\ln Z$ appears in Eq. 3 with a negative sign, estimating it with a suitable (i.e., low-variance) upper bound would lead to an overall lower bound on the log marginal likelihood, which one can then maximize as usual, hopefully with fewer optimization issues.

## Minor Nitpicks

- acronym VLB for variational lower bound is used without being defined (i.e., "(VLB)" missing at the end of line 80).
- the introduction reads more like a "Related Works" section.
- This might be a matter of taste, but I would no longer introduce VAEs as a "cornerstone in generative modeling" in the age of diffusion models. I'd rather highlight their capabilities for representation learning (and maybe data compression).

## References

- [1] [Bamler et al., Perturbative Black Box Variational Inference, NeurIPS 2017](https://proceedings.neurips.cc/paper/2017/hash/b75bd27b5a48a1b48987a18d831f6336-Abstract.html)
- [2] [Minka, Divergence measures and message passing, 2005](https://miat.inrae.fr/AIGM/biblios/TR-2005-173.pdf)
- [3] [Domke and Sheldon, Importance Weighting and Variational Inference, NeurIPS 2018](https://proceedings.neurips.cc/paper_files/paper/2018/file/25db67c5657914454081c6a18e93d6dd-Paper.pdf)

**Questions:**

- How big is the variance of the gradient of the estimator of $Z$? Can it be reduced using similar ideas as in VI? And would this variance possibly explain the optimization problems?
- How is the second term (the "$-\ln Z$"-part) of Eq. 3 estimated, and is my assumption correct that the expectation of its estimator is an upper rather than a lower bound on $(-\ln Z)$?

---

### Note · Authors · 2024-11-25

I have read and agree with the venue's withdrawal policy on behalf of myself and my co-authors.